# Diversity and Composition of Fungicolous Fungi Residing in Macrofungi from the Qinling Mountains

**DOI:** 10.3390/jof10090601

**Published:** 2024-08-25

**Authors:** Wenyan Huo, Langjun Cui, Pengdong Yan, Xuelian He, Liguang Zhang, Yu Liu, Lu Dai, Peng Qi, Suying Hu, Ting Qiao, Junzhi Li

**Affiliations:** 1Fungal Research Center, Shaanxi Provincial Institute of Microbiology, Xi’an 710043, China; huowenyan0616@126.com (W.H.); ly137261323@126.com (Y.L.);; 2College of Life Science, Shaanxi Normal University, Xi’an 710062, China

**Keywords:** diversity, fungicolous fungi, macrofungal fungi, Qinling Mountains

## Abstract

Sporocarps of macrofungi support other diverse fungal species that are termed fungicolous fungi. However, the external environmental factors that affect the diversity and composition of fungicolous fungal communities remains largely unknown. In this study, the diversities, composition, and trophic modes of fungicolous fungal communities residing in host macrofungi from diverse habitats in the Qinling Mountains were analyzed. Additionally, the number of carbohydrate-active enzymes (CAZymes) encoded by saprophytic, pathogenic, and symbiotic fungi was also quantified and compared. The results revealed that the diversity and composition of fungicolous fungal communities varied with months of collection and the habitats of host fungi, and saprophytic fungi were more abundant on wood than on the ground. Meanwhile, it was also found that saprophytic fungi possessed higher abundances of cell-wall-degrading enzymes than pathogenic or symbiotic fungi. Based on the above findings, it was hypothesized that the greater abundance of saprophytic fungi on wood compared to the ground may be due to their possession of a more diverse array of enzymes capable of degrading wood cell walls, thereby allowing for more efficient nutrient acquisition from decaying wood.

## 1. Introduction

Sporocarps of macrofungi support other diverse species, and especially other microorganisms [1,2]. Moreover, the growth and development of macrofungi fruiting bodies are intimately associated with these other microorganisms [3]. The roles of microorganisms inhabiting fruiting bodies have been extensively investigated [3]. For example, *Pseudomonas putida* was isolated from the sporocarp of *Agaricus bisporus* as early as 1991 and was found to promote the growth of the host mycelia but inhibit the hyphal branching frequency [3,4]. Oh et al. [5] recently demonstrated that isolates of *Dietzia*, *Ewingella*, *Pseudomonas*, *Paenibacillus*, and *Rodococcus* enhanced the growth of the host pine mushroom (Tricholoma matsutake), while other bacterial isolates including *Mycetocola* and *Stenotrophomona* negatively affected host growth due to the activities of their various enzymes including chitinases, cellulases, and proteases [3,5]. Furthermore, Ma et al. revealed that *Pseudomonas* was the predominant bacterial genus associated with the sporocarp of the host *Shiraia* and promoted greater production of the photosensitive drug hypocrellin A by the host [6].

Fungi that feed on other fungi are termed fungicolous [7,8]. Fungicolous fungi diversity, occurrence, and interactions with host fungi remain poorly known [9]. Notably, lower fungal diversity and richness have been reported inside the sporocarp of macrofungi. Approximately 1,500 fungicolous taxa have been identified [10]. Given that there are an estimated 2 to 4 million fungal species globally [11], fungicolous fungal diversity is likely to be grossly underestimated [9]. Most fungicolous fungi do not generate sporocarp, which may be why they are largely overlooked, despite their wide distribution [9,12]. Yeasts comprise one of these neglected groups and are also identified as endophytes of plants [13,14] and lichens [15], in addition to participation in tripartite interactions with parasitic fungi and mushrooms [16]. Despite some taxonomic studies [17,18,19,20] and investigations of parasitism by fungicolous fungi [10], few studies have evaluated the diversity and ecological roles of the fungicolous fungi of macrofungi. A recent investigation of fungal communities residing within sporocarps of various fungal hosts revealed that a significant proportion of fungicolous fungi was poorly represented in reference databases, suggesting insufficient documentation of their diversity [9,21]. Furthermore, the study indicated that fungicolous fungi were represented across widespread different lineages within the fungal kingdom, although most were in the Ascomycota and most exhibited unknown ecological functions [9,21]. Nevertheless, many questions remain about fungicolous fungi, including what the environmental drivers of fungicolous fungal community composition are and whether macrofungi growing on different substrates harbor different diversity and community compositions of fungicolous fungi.

One reason the aforementioned questions remain unsolved is that investigations of fungicolous fungi have primarily relied on cultivation and morphological analyses that only provide limited insight into their diversity [22]. DNA metabarcoding has been used as a powerful tool to investigate fungal diversity and is based on high-throughput sequencing (HTS) of amplified marker gene(s). Importantly, the composition of the community does not need to be known in advance [9]. Recently, Koskinen et al. used DNA metabarcoding to document a high diversity of fungicolous fungi in soft and short-lived agarics (Agaricus, Boletales, and Rusulales), although no apparent host preference was identified. The apparent lack of host specificity could be partially explained by the relatively short lifespans of fleshy agarics compared with other annual fungi [1].

Here, the main aim of this study was to analyze the diversity, community composition, as well as trophic mode diversity of fungicolous fungi residing in host macrofungi from the Qinling Mountains based on DNA metabarcoding. To the best of our knowledge, this is the first attempt to investigate the diversity and composition of fungicolous fungi residing in macrofungi from the Qinling Mountains and contributes to a better understanding of the diversity and function of fungicolous fungi. It was found that the diversity and composition of fungicolous fungal communities varied with the months of collection and the habitats of host fungi. This study further revealed that the abundance of saprophytic fungi on wood was significantly higher than on the ground, and that saprophytic fungi possessed higher abundances of cell-wall-degrading enzymes. Based on the above findings, a hypothesis has been proposed to explain why there is a higher proportion of saprophytic fungi on wood than on the ground.

This study provides an analysis of the diversity and environmental interactions of fungicolous fungal communities within macrofungi. The findings suggest that the timing of host growth and the specific habitat are pivotal in determining the diversity and composition of these communities. The observed differences in abundance of CWDES among fungal types offer insights into the ecological roles and competitive advantages of saprophytic fungi in forest ecosystems.

## 2. Materials and Methods

### 2.1. Sample Collection

Samples were collected from the Qinling Mountain within the Shaanxi Province (China) from March 2020 to November 2021. The sampling regions encompassed a range of areas, including Ningshan, Zhashui, Mei, Foping, Taibai, and Chang’an districts. The survey extended from its lowest elevation at Longxu Gully in Xialiang Town, Zhashui County (at 608 m) to its highest in the Pinghe Liang area, Ningshan County (at 2364 m), showcasing a substantial altitudinal range of 1756 m. The most southerly sampling points were near Ningshan County, Shaanxi Province (33.38 N, 108.26 E), and the northernmost was at Lintong District, Shaanxi Province (34.34 N, 109.26 E), while the most easterly sampling point was at Lintong District, Shaanxi Province (34.34 N, 109.26 E), and the westernmost was near Taibai County, Shaanxi Province (34.15 N, 107.26 E). 

The sampling region is located within the climatic transition belt from subtropical to warm temperate zones, marking it as one of the primary areas rich in macrofungal diversity in the Qinling Mountains. The vegetation in this region comprises various forest ecosystems, including deciduous broadleaf forests, mixed coniferous and broadleaf forests, coniferous forests, and scrub forests. The deciduous broadleaf forests are dominated by species such as *Quercus variabilis*, *Quercus acutissima*, and *Quercus wutaishanica*. The mixed coniferous and broadleaf forests feature *Pinus armandii*, *Quercus wutaishanica*, and *Quercus acutissima* as their principal tree species. The coniferous forests are characterized predominantly by *Pinus tabulaeformis*, *Pinus armandii*, and *Metasequoia glyptostroboides*. Additionally, the scrub forests are notably dominated by *Fargesia spathacea*.

To avoid the pseudo-replication effect, a minimum elevation gradient of over 100 m was maintained across all sampling sites. To minimize the influence of factors such as humidity on the experimental outcomes and to ensure the acquisition of a sufficient quantity of samples, the collection was scheduled within the 1 to 3 days following rainfall at the pre-designated sampling sites.

The specimens were preliminarily identified in the field based on fruiting body morphology. To identify the specimens more accurately, the microscopic characteristics of fruiting bodies were observed using a field light microscope (Olympus BX43, Olympus, Tokyo, Japan), alongside molecular identification based on ITS DNA sequencing. To avoid aerial contaminants, the outer surfaces of each sporocarp were removed from a total of 280 unique sporocarps (see Appendix A for detailed information). Between 10 and 15 sections of approximately 5 mm^3^ were removed from the subiculum layer and then placed in 2 mL tubes containing DNA-EZ Reagents F DNA-Be-Locked A (Sangon Biotech, Shanghai, China) for DNA extraction and metabarcoding.

### 2.2. DNA Extraction

The removed pieces of the 280 specimens were separately ground into powder with liquid nitrogen using a sterilized 12 cm mortar and pestle and then placed on ice. The total genomic DNAs of the associated macrofungi were subsequently extracted using the Ezup Column Fungi Genomic DNA Purification Kit (Sangon Biotech, Shanghai, China) according to the manufacturer’s protocol. The DNA integrity, purity (OD260/OD280 = 1.8–2.0), and concentration were measured using a OneDropTM OD-1000+ Spectrophotometer (Nanjing Wuyi Technology Co., Ltd., Nanjing, China). Extracted DNA was then used for the subsequent PCR amplification of the internal transcribed spacer (ITS) region and preparation of metabarcoding libraries.

### 2.3. PCR Amplification and Sequencing of Internal Transcribed Spacer Regions

PCRs targeting the ITS region were performed using the universal primers ITS4 (TCCTCCGCTTATTGATATGC) and ITS5 (GGAAGTAAAAGTCGTAACAAGG) [23]. PCRs with a total volume of 30 µL comprised 15 µL of 2× Rapid Taq Master Mix (Tsingke Biological Technology, Beijing, China), 1.0 μL each of the forward and reverse primers (10 mM), 1.5 µL of DNA template, and 11.5 μL of ddH_2_O. After initial denaturation at 95 °C for 3 min, amplification was performed over 30 cycles comprising 20 s at 95 °C, 20 s at 55 °C, and 70 s at 72 °C, followed by a final extension at 72 °C for 5 min. DNA sequencing was conducted at the Tsingke Biological Technology Co., Ltd. (Xi’an, China), using an ABI PRISM 3730XL Analyzer (Life Technologies, Gaithersburg, MD, USA) with the same primers used for PCR.

### 2.4. Preparation of Metabarcoding Libraries and Sequencing

A total of 16 metabarcoding libraries were constructed from the DNA extracts of 280 samples, each containing ITS1 sequences from 15 to 30 macrofungal samples. Amplicon libraries were constructed using combinations of 15–30 uniquely tagged primers designed to target the ITS1 region, with the modified reverse primer ITS2ngs (5′-xTTYRCKRCGTTCTTCATCG-3′) and modified forward primer ITS1catta (5′-xACCWGCGGARGGATCATTA-3′). Detailed information on barcodes is shown in Appendix A. PCRs were conducted in a final volume of 50 µL, comprising 25 µL of 2× Rapid Taq Master Mix (Tsingke Biological Technology, Beijing, China), 1.5 μL each of the forward and reverse primers (10 mM), 1.5 µL of DNA template (100 ng/µL), and 20.5 μL of ddH_2_O. The PCR amplification conditions included 95 °C for 3 min, followed by 10 cycles of 95 °C for 20 s, 47 °C for 20 s, and 72 °C for 30 s, in addition to 25 cycles of 95 °C for 20 s, 59 °C for 20 s, and 72 °C for 30 s, all followed by a final extension at 72 °C for 5 min. PCR products were evaluated using electrophoresis on a 1.5% agarose gel, followed by purification and pooling using the SequalPrep Normalisation Plate Kit (Invitrogen, San Diego, CA, USA). Between 15 and 30 PCR products were pooled, concentrated, and cleaned using Agencourt AMPure XP magnetic beads (Nerliens Meszansky AS, Oslo, Norway). Quality control was conducted using a dsDNA 1000 Bioanalyzer (Agilent Technologies, Santa Clara, CA, USA) and Qubit instrument (Life Technologies). Sequencing was conducted on the Illumina HiSeq platform at Tsingke Biological Technology Co., Ltd. (Xi’an, China), by generating 2 × 250-bp paired-end reads.

### 2.5. Bioinformatic Processing and Analysis of Sequencing Data

The QIIME 2 v2022.2 software was employed to construct a naive Bayes classifier using 196344 fungal ITS sequences from the Unite database [24,25].

Using the Illumina HiSeq platform, a dataset was generated that included 83,943,646 reads. These reads were filtered with the fastq tool [26], applying a sliding window length (-w) of 16, and no quality or length filtering was performed (using the options -disable_quality_filtering and -disable_length_filtering). Sequencing adapters were detected and removed using fastq with the detect_adapter_for_pe option. The reads of the pooled samples were split using their barcodes with the fastq-multx program (https://github.com/brwnj/fastq-multx, accessed on 11 October 2022). Barcode removal from reads was conducted using DADA2 [27]. Amplicon sequence variants (ASVs) from each sample, including ASVs of host species and fungicolous fungi, were also generated using DADA2 with the denoising method [27]. The trained classifier was used to annotate the ASVs. ASV information of host species, including the numbers of ASV sequences, the species annotation results, and annotation confidence were removed with custom Python scripts (Appendix A).

The annotation results of the classifier were subjected to alpha diversity (Chao1 and Shannon) analysis, non-metric multidimensional scaling analysis (NMDS), and linear discriminant analysis (LDA) effect size (LEfSe) calculations using the MicrobiomeAnalyst platform (https://www.microbiomeanalyst.ca/, accessed on 11 October 2022). In addition, the Fungi Functional Guild (FUNGuild, http://www.funguild.org/, accessed on 11 October 2022) platform was used to identify the functional groups of hosts and fungicolous fungi [28]. Briefly, the host fungi and all fungicolous fungal ASV taxonomic annotations at the genus or family levels were submitted to FUNGuild and then broadly classified based on the trophic modes of “pathotrophs”, “saprotrophs”, “symbiotrophs”, and “others” using highly probable confidence values and assignments according to the primary literature or authoritative resources [28]; “others” refers to fungal species that cannot be classified into a single trophic mode. The annotated protein sequences from 216 fungi were retrieved from NCBI (https://www.ncbi.nlm.nih.gov/, accessed on 11 October 2022) and used to identify carbohydrate-active enzymes (CAZymes) with the dbCAN platform [29] based on Diamond, HMMER, and Hotpep models. During annotation, the protein sequence features were used as search targets, and the files containing all fungal protein sequences were used as search objects. The preliminary results were processed using a custom python script (Appendix A) to count the number of various CAZymes for each fungus.

### 2.6. Compilation of Metadata

The samples were classified into different categories according to different environmental parameters. Specifically, six categories based on the collection months of the host species were used: “March–April” (39 samples), “May–June” (36 samples), “July” (47 Samples), “August” (61 samples), “September” (82 samples), and “October–November” (15 samples). The samples were also divided into growth habitat categories including “coniferous forests” (30 samples), “hardwood forests” (or “deciduous broadleaf forests”, 109 samples), “mixed forests” (131 samples), and “scrub forests” (10 samples). In addition, host macrofungi were divided into altitude categories of “intermediate” (179 samples, collected from areas with an altitude < 1500 m) and “high” (101 samples, collected from areas with an altitude of >1500 m), in addition to growth substrate categories of “wood” (111 samples) and “ground” (169 samples). 

### 2.7. Statistical Analyses

All data are presented as means ± standard deviations (SD). Differences between groups were analyzed using Student’s *t*-tests or one-way ANOVA tests using the GraphPad Prism 5.0 software. Differences were considered statistically significant if *p* < 0.05.

## 3. Results

### 3.1. Identification of Collected Macrofungi

Identification of the 280 fungal samples was conducted based on morphological features of the fruiting bodies, with reference to microscopic features and/or sequence information from the ITS region if necessary. The samples belonged to 186 species, 100 genera, 53 families, 18 orders, 7 classes, and 2 phyla (Appendix A). In addition, four samples comprised novel species that have been officially described [30,31,32] (Appendix A and Figure 1). Furthermore, 17 samples comprised novel species that require further validation (Appendix A and Figure 1).

### 3.2. Distribution of Fungicolous Fungi in Macrofungal Samples

The diversity of fungicolous fungi inhabiting sporocarps of 280 fungal host species collected from the Qinling Mountains was evaluated using a metabarcoding approach. A total of 769 species of fungicolous fungi were identified belonging to 573 genera, 225 families, 99 orders, 36 classes, and 15 phyla. Fungicolous fungal species and genera with relative abundances < 1% were excluded. Among fungicolous fungi with relative abundances > 1% in each sample, the most widely distributed species were *Simplicillium cylindrosporum*, followed by *Tyromyces chioneus*, *Stereum hirsutum*, *Laccaria amethystina*, and *Neofavolus alveolaris*, which were identified in 21.07%, 17.50%, 12.14%, 11.79%, and 11.43% of the samples, respectively (see Figure 2a). The most widely distributed genera were *Mycena*, *Gymnopus*, *Psathyrella*, *Simplicillium*, and *Russula*, which were found in 43.57%, 28.57%, 23.57%, 21.07%, and 20.36% of the samples, respectively (see Figure 2b).

### 3.3. Diversity and Composition of Fungicolous Fungal Communities Varied with Months of Collection and the Habitats of Host Fungi

To identify the environmental variables that were associated with the diversity and composition of fungicolous fungi, the alpha and beta diversities of fungicolous fungal communities were analyzed and compared between hosts growing in different environments, from different collection months, habitats, altitudes, and growth substrates. Significant differences in Chao1 and Shannon index values were observed between samples collected in different months or from different habitats (Figure 2). Host fungi collected in August exhibited higher diversity (for both Chao1 and Shannon index values) than those collected in other months (Figure 2c,d, *p* < 0.001). Hosts from coniferous forests also exhibited higher fungicolous fungal diversity than fungi from other habitats (Figure 2e,f, *p* < 0.001). In addition, small differences were observed for both the Chao1 and Shannon index values of fungicolous fungi for host species from different altitude areas or growth substrates (Appendix A).

Community compositional variation in fungicolous fungi was evaluated between samples of different categories using NMDS analysis based on Bray–Curtis dissimilarities of fungicolous fungal communities (Figure 2 and Appendix A). Each sample was distinct from the others, and a high degree of variation was observed between samples. Moreover, significant differences in fungicolous fungal community compositions were observed between the macrofungi sampled in different months or from different habitats (Figure 2, *p* = 0.001). However, these differences were not clear for host macrofungi collected from different altitudes and with different growth substrates (Appendix A and Appendix A). These results were overall consistent with Shannon and Chao1 index value variation.

LEfSe was used to identify individual fungicolous fungi species or genera that exhibited statistically different relative abundances between host fungi sampled in different months or from different habitats. A total of 20 species and 22 genera exhibited significantly different enrichment among sampling months (Appendix A), while 18 species and 15 genera exhibited significantly different enrichment between samples from different habitats (Appendix A).

The distribution of fungicolous fungi in host species collected in different months or from different habitats was also evaluated. The most widely distributed fungicolous fungal species were *Tubaria praestans* (March–April), *Laccaria amethystina* (May–June), *Simplicillium cylindrosporum* (July), *Agaricus moelleroides* (August), *Stereum hirsutum* (September), and *Hypholoma fasciculare* (October–November), which were identified in 56.41%, 50.00%, 38.30%, 39.34%, 30.49%, and 40.00% of samples, respectively (Table 1). The most widely distributed genera were *Psathyrella* (March–April), *Entoloma* (May–June), *Russula* (July), *Mycena* (August), *Mycena* (September), and *Gymnopus* (October–November), which were identified in 61.54%, 58.33%, 53.19%, 63.93%, 43.90%, and 73.33% of host macrofungi, respectively (Table 1). Among the fungicolous fungi with relative abundances > 1% in each sample, the most widely distributed species in host samples collected from the scrub, mixed, hardwood, and coniferous forest habitats were *Tubaria praestans*, *Stereum hirsutum*, *Tyromyces chioneus*, and *Agaricus moelleroides*, which were identified in 70.00%, 22.90%, 35.78%, and 43.33% of hosts, respectively (Table 2). At the genus level, the most widely distributed fungicolous fungi were *Tubaria*, *Mycena*, *Mycena*, and *Agaricus*, which were identified in 70.00%, 32.06%, 52.30%, and 76.67% of host species, respectively (Table 2). Taken together, these results indicate that the diversity and composition of fungicolous fungi varied with the months of collection and the habitats of host fungi.

### 3.4. Trophic Modes of Host Macrofungi and Fungicolous Fungi

The lifestyle of fungicolous fungi were categorized into three trophic modes and compared. The proportion of saprophytic fungicolous fungi in samples decreased on average from March to July, while the proportion of symbiotic fungi gradually increased (*p* < 0.0001, Table 3 and Figure 3). The proportion of saprophytic fungi gradually increased from July to November, while the proportion of symbiotic fungi decreased over the same period (*p* < 0.0001, Table 3 and Figure 3). Compared with host macrofungi growing in other forests, the host macrofungi growing in mixed forests hosted fewer saprophytic fungicolous fungi and more symbiotic fungi (*p* < 0.0001, Table 3 and Figure 3). In addition, host species growing on wood harbored more saprophytic fungicolous fungi and fewer symbiotic fungi compared with the hosts growing on the ground (*p* < 0.0001, Table 3 and Figure 3).

The trophic modes of macrofungal hosts were also enumerated. Among the 280 hosts, 147 were saprophytic fungi, 2 were pathotrophs, and 75 were symbiotrophs (Appendix A). Among the 111 host macrofungi growing on wood, 79 were saprophytic, accounting for 71.2% of the total, while only 5 were symbiotic (4.5% of the total; Appendix A). Among the 169 hosts growing on the ground, 68 were saprophytic and 70 were symbiotic, accounting for 40.2% and 41.4% of the totals, respectively (Appendix A).

### 3.5. Carbohydrate-Active Enzyme Diversity of Host Macrofungi and Fungicolous Fungi

The annotated protein sequences of 216 host or fungicolous fungi, comprising 96 saprophytic fungi, 58 parasitic fungi, and 62 symbiotic fungi, were annotated against the CAZyme database using the dbCAN platform. The results from all three models were consistent. Overall, saprophytic fungi possessed higher abundances of CAZymes than parasitic and symbiotic fungi (*p* < 0.0001, Figure 4). Saprophytic fungi encoded greater abundances of glycoside hydrolases (GHs), auxiliary activities (AAs), carbohydrate esterases (CEs), polysaccharide lyases (PLs), and carbohydrate-binding modules (CBMs), as well as fewer glycosyltransferases (GTs) (*p* < 0.05, Figure 4).

## 4. Discussion

HTS studies have reshaped our understanding of fungi, revealing enormous and largely unstudied taxonomic and functional diversity [33]. Many software pipelines are available for processing and analyzing HTS metabarcoding data, with the mothur (https://www.mothur.org, accessed on 15 October 2022) [34], USEARCH (https://www.drive5.com/usearch, accessed on 15 October 2022) [35], and QIIME (https://qiime2.org, accessed on 15 October 2022) [36] platforms being among the most common [33]. Three primary reference ITS databases are currently available for fungi, namely the INSDC, UNITE [24], and Warcup databases [33,37]. 

In this study, based on the Unite database, a software pipeline has been developed specifically for the simultaneous processing and analysis of reads from both hosts and their associated fungicolous fungi that can derive from the sporocarps of macrofungi. The pipeline comprises six steps. (I) PCR products with different barcodes for samples were mixed, and sequencing was performed on an Illumina HiSeq platform with 2 × 250-bp paired-end reads to obtain sequence reads. (II) Then, fastq was used to filter reads, in addition to detecting and removing adapters. (III) Reads of the mixed samples were then split using the fastq-multx toolkit based on barcodes. (IV) After the removal of the barcodes, ASVs of the host species and fungicolous fungi were obtained from the denoising pipeline of DADA2. (V) Then, classifiers were trained based on Unite database and used to annotate fungi ASVs. (VI) Lastly, the non-host ASVs were used for further analysis and to identify the diversity and composition of fungicolous fungi (Appendix A).

Although high-throughput sequencing technologies such as Illumina amplicon sequencing provide us with powerful tools for identifying the diversity of fungi in macrofungi, they have limitations in determining the specific life stages of these fungi within the fruiting bodies (such as mycelium, spores, or resistance structures). Future research may combine traditional microscopic observation and cultivation techniques to gain a more comprehensive understanding of the life cycle of these fungi.

Previous studies have shown that fungicolous fungi from different fungal lineages were widespread within macrofungal sporocarps [21,33]. Fungicolous fungi were detected in all of the 280 samples, consistent with the results of previous studies [21,33]. Sporocarps of macrofungi can act as small islands among a forest landscape, providing habitat and resources for a larger pool of fungicolous fungi [9]. Therefore, the results of this study suggest that macrofungal sporocarps are not just individual units, but rather are complex communities comprising various microorganisms. Fungicolous fungal diversity within these complex communities has been shown to vary with the life-history traits of the host fungi like sporocarp lifespan or morphology [9]. For example, short-lived sporocarps hosted more fungicolous fungal diversity based on OTU richness and Shannon diversity, while resupinate sporocarps harbored a greater diversity of fungicolous fungi than pileate species [9]. However, the effects of external environmental variables, such as month of collection, habitat, substrates, and altitudes of the host, on the diversity and composition of fungicolous fungi have been little studied. Here, the alpha and beta diversities of fungicolous fungal communities were analyzed and compared between macrofungi from different environments to evaluate the environmental variables that most significantly influence the diversity and composition of fungicolous fungi. The results indicate that the diversity and composition of fungicolous fungal communities varied with the months of collection and the habitats of the host fungi (Figure 2).

Fungicolous fungi play significant evolutionary and ecological roles within their ecosystems [10]. For instance, fungicolous fungi contribute to the decomposition process, helping to break down organic matter and recycle nutrients within ecosystems [10]; the interactions between fungicolous fungi and their hosts can drive biodiversity and evolution, as seen in the diverse adaptations and strategies employed by these organisms to coexist or outcompete each other [10]. Fungicolous fungi can also affect the cultivation of edible and medicinal mushrooms by competing for resources or by causing diseases that reduce yield or quality [10]. Future research should explore the potential applications of these fungi in the field of biological control, as well as investigate how they influence the productivity and quality of cultivated mushrooms.

CAZymes are a large class of important enzymes that can be separated by function into six categories, namely GTs, GHs, AAs, CEs, PLs, and CBMs [38,39]. CAZymes involved in carbohydrate degradation include GHs, PLs, and CEs [40] that are also known as cell-wall-degrading enzymes (CWDES) [41]. AAs perform various redox conversions that are often intrinsically related to degradation [40]. GTs are primarily involved in carbohydrate synthesis, while CBMs assist enzymes in recognizing and binding specific substrates [40]. Fungal CWDES have recently received considerable research attention. CWDES can degrade plant cell walls and release the sugar monomers within them, thereby providing nutrients for fungal growth [41,42]. Depending on their nutrient sources (i.e., trophic modes), fungi can be divided into saprophytic, symbiotic, and parasitic groups. Saprophytic fungi primarily rely on CWDES to degrade plant cell walls and release the sugars within them as nutrients to meet their growth and reproduction needs [42]. In this study, host species growing on wood harbored more saprophytic fungicolous fungi than hosts growing on the ground, while the host macrofungi growing on wood were mostly saprophytic fungi. Based on these results and previous studies, saprophytic fungi are hypothesized to produce greater carbohydrate-degrading enzymes, and especially CWDES, that can degrade plant cell walls and release the sugars within them, thereby providing nutrients for the growth and reproduction of mycelia. Thus, wood is more suitable for the growth of saprophytic fungal mycelia than the ground, leading to a higher abundance of saprophytic fungal mycelia in them. That is to say that wood exerts an enriching influence on the mycelium of saprophytic fungi. Under suitable environmental conditions, these saprophytic fungal mycelia continuously propagate and some of them form sporocarps. Other fungal mycelia may also take the opportunity to “infiltrate” these formed sporocarps. From the perspective of nutrient utilization, this hypothesis partially explains the formation mechanism of the “small islands” growing on wood and can serve as a supplement to the “small islands” theory (Figure 5) [9].

If the above hypothesis is true, saprophytic fungi are likely to host more CAZymes, and especially CWDES, compared with parasitic and symbiotic fungi, regardless of whether we consider host or fungicolous fungi. Consistent with this hypothesis, a statistical analysis of the CAZyme abundances in 216 fungi, comprising 96 saprophytic fungi, 58 parasitic fungi, and 62 symbiotic fungi, revealed that saprophytic fungi harbored a higher proportion of CAZymes, and especially CWDES. Thus, the above hypothesis is supported by the results of this study.

## 5. Conclusions

Here, the alpha and beta diversities of fungicolous fungal communities residing in host macrofungi were analyzed, revealing that the diversity and composition of fungicolous fungi varied with the time of collection and habitat of the host fungi. It was further revealed that saprophytic fungi were significantly more abundant on wood than on the ground. Meanwhile, it was also found that saprophytic fungi possessed higher abundances of cell-wall-degrading enzymes. Based on the above findings, it was hypothesized that the higher content of saprophytic fungi in wood may be attributed to their possession of a greater number of cell-wall-degrading enzymes, which enabled the more efficient utilization of nutrients released during the decay of wood. These results not only provide critical insights for the in-depth understanding of macrofungi and their ecosystems, but also provide important theoretical and practical guidance to promote the sustainable development and utilization of fungal resources.

## Figures and Tables

**Figure 1 jof-10-00601-f001:**
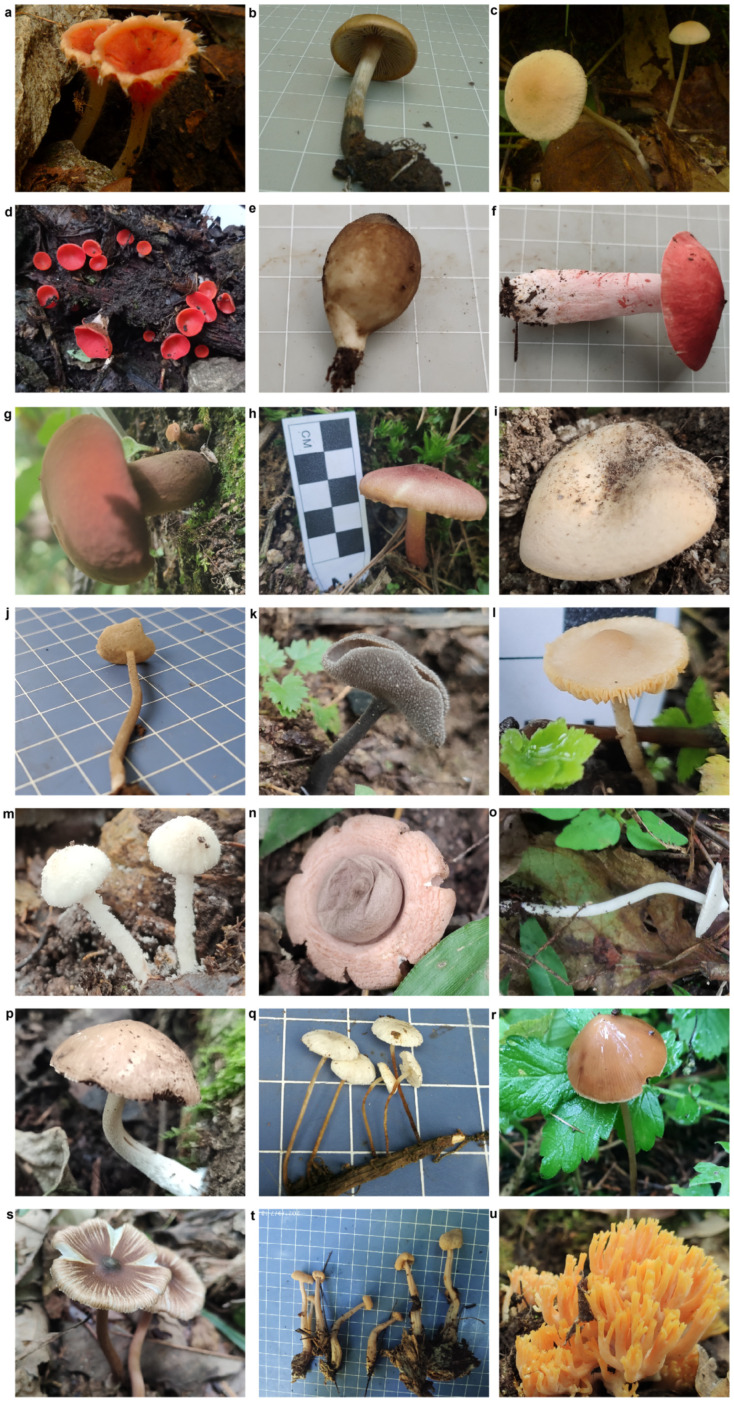
Novel fungal species identified in this study. Published novel species of *Microstoma* (**a**), *Psilocybe* (**b**), and *Helvella* (**j**,**k**). Novel species of *Marasmius* (**c**), *Sarcoscypha* (**d**), *Lycoperdon* (**e**), *Russula* (**f**), *Boletaceae* (**g**), *Tricholomopsis* (**h**), *Lactifluus* (**i**), *Tubaria* (**l**), *Cystolepiota* (**m**), *Geastrum* (**n**), *Inocybe* (**o**,**s**), *Psathyrella* (**p**), *Marasmiellus* (**q**), *Conocybe* (**r**), *Cudonia* (**t**), and *Ramaria* (**u**).

**Figure 2 jof-10-00601-f002:**
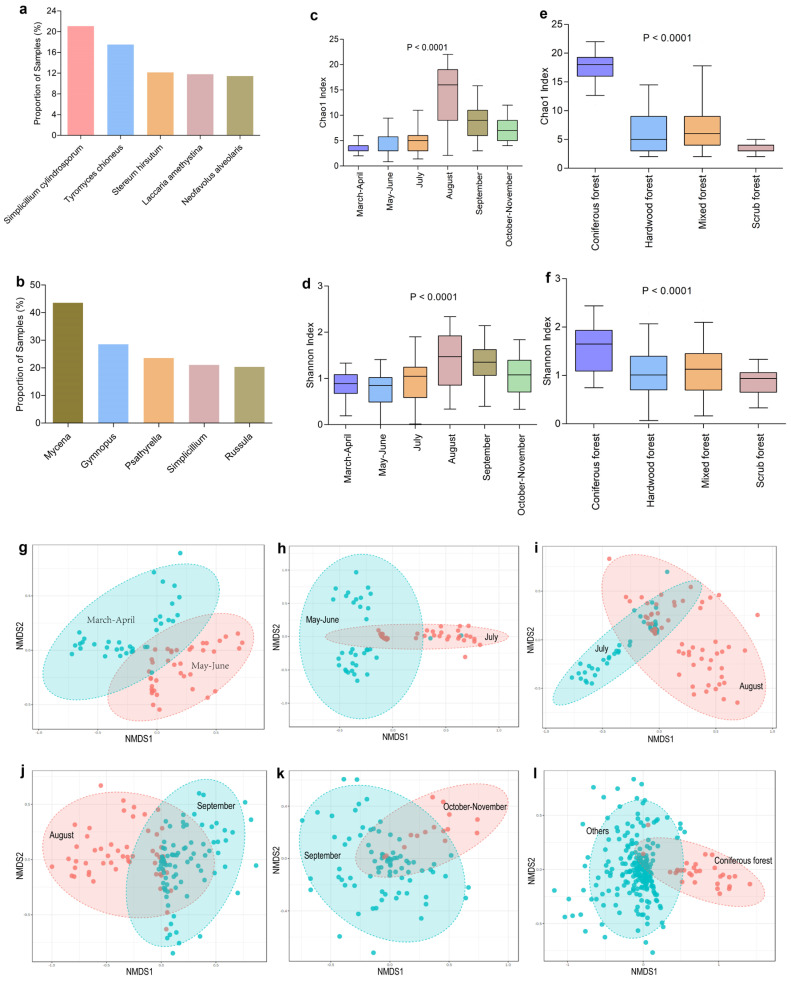
Diversity and composition of fungicolous fungal communities varied with months of collection and the habitats of host fungi. The most widely distributed fungicolous fungi at the species (**a**) and genus (**b**) levels in macrofungal samples. Alpha diversity of fungicolous fungal communities (**c**–**f**); box plots showing variation in Chao1 (**c**,**e**) and Shannon diversity (**d**,**f**) values of fungicolous fungi of sporocarps sampled in different months (**c**,**d**) and from different habitats (**e**,**f**); medians are indicated by dark lines; statistical differences between the different categories were evaluated using one-way ANOVA tests. Non-metric multidimensional scaling (NMDS) plots are shown based on Bray–Curtis dissimilarities of fungicolous fungal community composition in sporocarps sampled in different months (**g**–**k**) and from different habitats (coniferous forests and others) (**l**); NMDS stress values were 0.17056 (**g**), 0.096494 (**h**), 0.12495 (**i**), 0.1652 (**j**), 0.18962 (**k**), and 0.19245 (**l**); statistical significance was based on PERMANOVA tests, and all *p* values were ≤0.001. Source data are provided as a source data file (Appendix A).

**Figure 3 jof-10-00601-f003:**
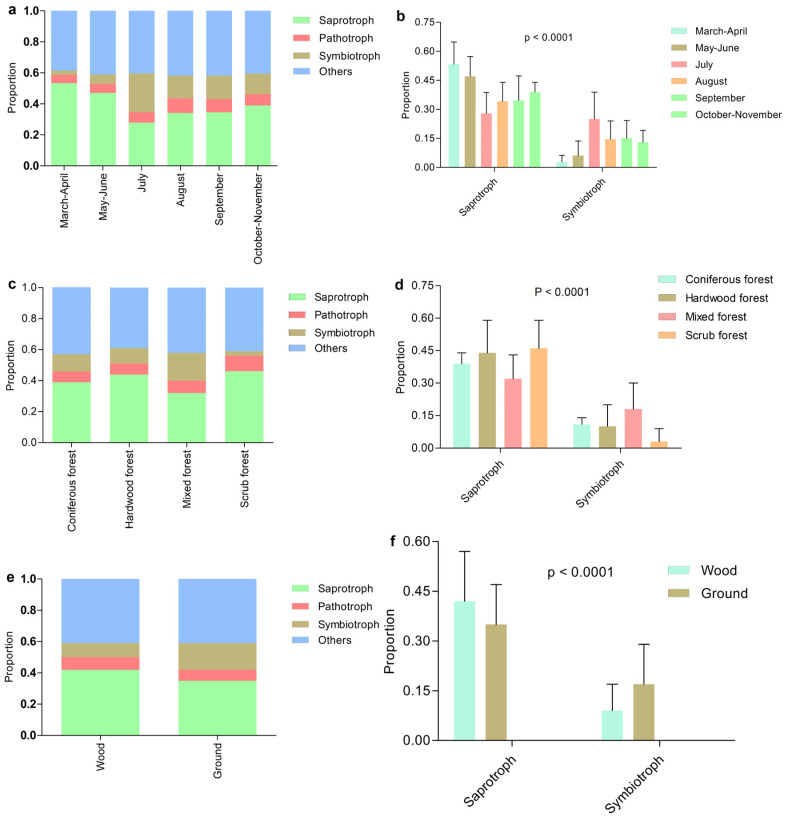
Comparisons of fungicolous fungi trophic modes. Fungicolous fungi from different host macrofungi with different collection months (**a**,**b**), habitats (**c**,**d**), and substrates (**e**,**f**). Statistical significance was analyzed using Student’s *t*-tests (**e**,**f**) or one-way ANOVA tests (**a**–**d**). Source data are provided as a source data file (Appendix A).

**Figure 4 jof-10-00601-f004:**
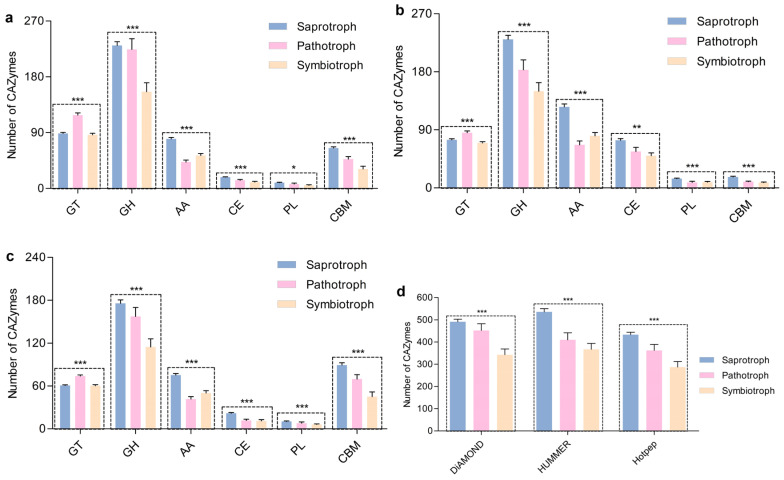
Number of different carbohydrate-active enzymes (CAZymes) encoded by saprophytic, pathogenic, and symbiotic fungi (**a**–**c**); CAZymes were predicted using DIAMOND (**a**), HUMMER (**b**), and Hotpep (**c**) models. Total number of CAZymes (**d**). Statistical significance was analyzed using one-way ANOVA tests. ***: *p* < 0.001, **: *p* < 0.01, and *: *p* < 0.05. GH: glycoside hydrolase; CBM: carbohydrate-binding module; CE: carbohydrate esterase; GT: glycosyltransferase; PL: polysaccharide lyase; AA: auxiliary activity. Source data are provided as a source data file (Appendix A).

**Figure 5 jof-10-00601-f005:**
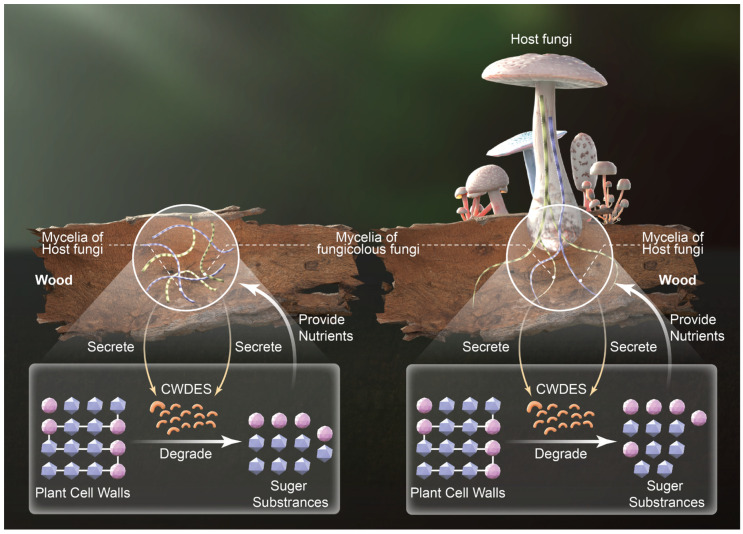
The hypothesis explaining why there is a higher proportion of saprophytic fungi on wood than on the ground.

**Table 1 jof-10-00601-t001:** Most widely distributed fungicolous fungal species and genera in macrofungal samples collected in different months.

Months	Widely Distributed Species		Widely Distributed Genera	
	Name	Proportion of Samples (%)	Name	Proportion of Samples (%)
March-April	*Tubaria praestans*	56.41	*Psathyrella*	61.54
May-June	*Laccaria amethystina*	50.00	*Entoloma*	58.33
July	*Simplicillium cylindrosporum*	38.30	*Russula*	53.19
August	*Agaricus moelleroides*	39.34	*Mycena*	63.93
September	*Stereum hirsutum*	30.49	*Mycena*	43.90
October-November	*Hypholoma fasciculare*	40.00	*Gymnopus*	73.33

Source data are provided as a source data file (Appendix A).

**Table 2 jof-10-00601-t002:** Most widely distributed fungicolous fungal species and genera in macrofungal samples collected from different habitats.

Habitat	Widely Distributed Species		Widely Distributed Genera	
	Name	Proportion of Samples (%)	Name	Proportion of Samples (%)
Scrub forest	*Tubaria praestans*	70.00	*Tubaria*	70.00
Mixed forest	*Stereum hirsutum*	22.90	*Mycena*	32.06
Hardwood forest	*Tyromyces chioneus*	35.78	*Mycena*	52.30
Coniferous forest	*Agaricus moelleroides*	76.67	*Agaricus*	76.67

Source data are provided as a source data file (Appendix A).

**Table 3 jof-10-00601-t003:** Trophic modes of fungicolous fungi from different sample types.

Categories		Saprotroph	Pathotroph	Symbiotroph	Others
Month	March–April	0.53 ± 0.11 ***	0.05 ± 0.05 *	0.03 ± 0.03 ***	0.39 ± 0.08
	May–June	0.47 ± 0.1 ***	0.06 ± 0.05 *	0.06 ± 0.07 ***	0.41 ± 0.08
	July	0.28 ± 0.11 ***	0.07 ± 0.06 *	0.25 ± 0.14 ***	0.4 ± 0.09
	August	0.34 ± 0.1 ***	0.09 ± 0.08 *	0.15 ± 0.09 ***	0.42 ± 0.08
	September	0.35 ± 0.13 ***	0.08 ± 0.07 *	0.15 ± 0.09 ***	0.42 ± 0.09
	October–November	0.39 ± 0.05 ***	0.07 ± 0.04 *	0.13 ± 0.06 ***	0.41 ± 0.06
Habitat	Coniferous forest	0.39 ± 0.05 ***	0.07 ± 0.03	0.11 ± 0.03 ***	0.44 ± 0.06 *
	Hardwood forest	0.44 ± 0.15 ***	0.07 ± 0.07	0.1 ± 0.1 ***	0.39 ± 0.09 *
	Mingled forest	0.32 ± 0.11 ***	0.08 ± 0.07	0.18 ± 0.12 ***	0.42 ± 0.08 *
	Scrub forest	0.46 ± 0.13 ***	0.1 ± 0.09	0.03 ± 0.06 ***	0.41 ± 0.06 *
Substrate	Wood	0.42 ± 0.15 ***	0.08 ± 0.08	0.09 ± 0.08 ***	0.41 ± 0.1
	Ground	0.35 ± 0.12 ***	0.07 ± 0.05	0.17 ± 0.12 ***	0.41 ± 0.08
Altitude	Intermediate	0.38 ± 0.15	0.08 ± 0.07	0.14 ± 0.13	0.41 ± 0.09
	High	0.38 ± 0.11	0.07 ± 0.05	0.13 ± 0.08	0.41 ± 0.08

Data are the mean and average proportions of fungicolous fungi with different trophic modes at ASV level in all samples. Statistical significance was analyzed using Student’s *t*-tests (based on substrate and altitude) or one-way ANOVA tests (month and habitat). ***: *p* < 0.001; *: *p* < 0.05. Values are means ± SD. Source data are provided as a source data file (Appendix A).

## Data Availability

The raw ITS1 amplicon reads have been deposited to the NCBI SRA database under the BioProject ID of PRJNA1099458 (BioSample accession: SAMN40942292). The annotated protein sequences of 216 fungi used in this study are available in the Figshare database (https://doi.org/10.6084/m9.figshare.25592886, accessed on 4 May 2024).

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
