# Peer review of "Diversity and Composition of Fungicolous Fungi Residing in Macrofungi from the Qinling Mountains"

_jof, 2024, doi:10.3390/jof10090601_

Round 1

Reviewer 1 Report

This manuscript is a novel and original study that explores the diversity, composition, and trophic modes of fungicolous fungal communities residing in host macrofungi in the Qinling Mountains, China. Additionally, the authors provide valuable information related to the quantification and comparison of carbohydrate-active enzymes (CAZymes) encoded by saprophytic, pathogenic, and symbiotic fungi inhabiting macrofungi. The primary strength of the manuscript lies in the potential for fungicolous fungi to serve as model organisms, facilitating a better understanding of species interactions, fungal evolution and divergence, and fungicolous mechanisms.

The article is undoubtedly publishable; however, there are two main observations that need to be addressed:

i) Due to the employed methodology, it is impossible to ascertain the stage at which the identified fungi were present in the fruit bodies—whether as mycelium, spores, or resistance structures. Molecular detection of fungal species does not definitively indicate that the fungi are in a somatic form. The authors have studied species identifiable by DNA metabarcoding, but the stage of presence—somatic (as mycelium), reproductive (as spores), or resistance structures (e.g., sclerotia)—remains unclear. This limitation of the Illumina amplicon tool must be thoroughly discussed; otherwise, any discussion of their interesting and novel work is weakened.

ii) It would be highly beneficial for the authors to include a discussion on thffffde evolutionary and ecological roles that fungicolous fungi studied in this work may play in the ecosystems where they reside. This could encompass aspects such as how the turnaround time of certain nutrients in food webs is affected by the rapid life cycles of some fungal species; the influence on population dynamics and sizes of their hosts in terrestrial ecosystems; the potential of mycoparasites of pathogenic fungi as biocontrol agents; and how some fungicolous fungi may act as causal agents of serious diseases in edible and medicinal mushrooms, thereby decreasing both yield and quality.

Some minor details require attention:

- Authors should carefully check the correct spelling of species throughout the text. For example, in line 28, they wrote “Laccaria amethystine” when the correct name is “Laccaria amethystina.”

-The reference list needs careful checking. For instance, reference number 42 is missing.

-The method for determining fungal lifestyles into categories such as Saprotroph, Pathotroph, Symbiotroph, and others should be clarified, including the classification system used.

-In Table S1, if the species have already been described, they should not be referred to as “novel species.” If they have not been described, the text can remain as “novel species”; otherwise, the species names should be included throughout the manuscript.

-In Table S5, clarify the term “grown on ground.” Specify whether it means “grown on forest floor,” “grown on organic matter,” “mineral soil,” etc., for better clarity for the readers.

Author Response

Comments 1: Due to the employed methodology, it is impossible to ascertain the stage at which the identified fungi were present in the fruit bodies—whether as mycelium, spores, or resistance structures. Molecular detection of fungal species does not definitively indicate that the fungi are in a somatic form. The authors have studied species identifiable by DNA metabarcoding, but the stage of presence—somatic (as mycelium), reproductive (as spores), or resistance structures (e.g., sclerotia)—remains unclear. This limitation of the Illumina amplicon tool must be thoroughly discussed; otherwise, any discussion of their interesting and novel work is weakened.

Response 1: Thank you for pointing it out, we agree with this comment. Therefore, we have added the following sentence to discuss the limitations of Illumina amplicon sequencing technology in determining the life stages of fungi. See lines 423 to 428 of revised manuscript or the following paragraphs.

“Although high-throughput sequencing technologies such as Illumina amplicon sequencing provide us with powerful tools for identifying the diversity of fungi in macrofungi, they have limitations in determining the specific stages of these fungi within the fruiting bodies (such as mycelium, spores, or resistance structures). Future research may combine traditional microscopic observation and cultivation techniques to gain a more comprehensive understanding of the life cycle of these fungi.”

Comments 2: It would be highly beneficial for the authors to include a discussion on the evolutionary and ecological roles that fungicolous fungi studied in this work may play in the ecosystems where they reside. This could encompass aspects such as how the turnaround time of certain nutrients in food webs is affected by the rapid life cycles of some fungal species; the influence on population dynamics and sizes of their hosts in terrestrial ecosystems; the potential of mycoparasites of pathogenic fungi as biocontrol agents; and how some fungicolous fungi may act as causal agents of serious diseases in edible and medicinal mushrooms, thereby decreasing both yield and quality.

Response 2: Thank you for pointing it out, we agree with this comment. Therefore, we have added the following sentences to discuss the evolutionary and ecological roles of fungicolous fungi in the ecosystems where they reside. See lines 448 to 457 of revised manuscript or the following paragraphs.

“Fungicolous fungi play significant evolutionary and ecological roles within their ecosystems [10]. For instance, Fungicolous fungi contribute to the decomposition process, helping to break down organic matter and recycle nutrients within ecosystems [10]; the interactions between fungicolous fungi and their hosts can drive biodiversity and evolution, as seen in the diverse adaptations and strategies employed by these organisms to coexist or outcompete each other [10]; Fungicolous fungi can also affect the cultivation of edible and medicinal mushrooms by competing for resources or by causing diseases that reduce yield or quality [10]. Future research should explore the potential applications of these fungi in the field of biological control, as well as investigate how they influence the productivity and quality of cultivated mushrooms.”

Comments 3: The reference list needs careful checking. For instance, reference number 42 is missing.

Response 3: Thank you for pointing it out, we agree with this comment. Therefore, we have carefully checked the list of references and found that reference number 1 was missing. It has now been included. See lines 535 to 536 of revised manuscript or the following paragraphs.

“1. Koskinen J, et al. Finding flies in the mushroom soup: Host specificity of fungus‐associated communities revisited with a novel molecular method. Mol Ecol. 2019; doi: 10.1111/mec.14810.”

Comments 4: Authors should carefully check the correct spelling of species throughout the text. For example, in line 28, they wrote “Laccaria amethystine” when the correct name is “Laccaria amethystina.”

Response 4: Thank you for pointing it out, we agree with this comment. Therefore, we have carefully checked the spelling of species names throughout the text and have corrected them where necessary. See line 321 of revised manuscript.

Comments 5: The method for determining fungal lifestyles into categories such as Saprotroph, Pathotroph, Symbiotroph, and others should be clarified, including the classification system used.

Response 5: Thank you for pointing it out, we agree with this comment. Therefore, the method has been described. See lines 183 to 190 of revised manuscript or the following paragraphs..

“In addition, the Fungi Functional Guild (FUNGuild, http://www.stbates.org/guilds/app.php) platform was used to identify the functional groups of hosts and fungicolous fungi [28]. Briefly, the host fungi and all fungicolous fungal ASV taxonomic annotations at the genus or family levels were submitted to FUNGuild and then broadly classified based on the trophic modes of “pathotrophs,” “saprotrophs,”, “symbiotrophs” and “others” using highly probable confidence values and assignments according to primary literature or authoritative resources [28]. "others" refers to fungal species that cannot be classified into a single trophic mode.”

Comments 6: In Table S1, if the species have already been described, they should not be referred to as “novel species.” If they have not been described, the text can remain as “novel species”; otherwise, the species names should be included throughout the manuscript.

Response 6: Thank you for pointing it out, we agree with this comment. Therefore, the term "Novel species" has been changed to "Published novel species", and “Potential novel species” has been changed to “Novel species” in Table S1, Table S2 and revised manuscript. See lines 254 to 257, 263 to 264 of revised manuscript, and Table S1, S2 of supplementary files.

Comments 7: In Table S5, clarify the term “grown on ground.” Specify whether it means “grown on forest floor,” “grown on organic matter,” “mineral soil,” etc., for better clarity for the readers.

Response 7: Thank you for pointing it out, we agree with this comment. Therefore, the term "Grown on ground" has been clarified in Table S5. It specifically means "on the forest floor". See Table S5 of supplementary files.

Reviewer 2 Report

The study is well-prepared, readable, and interesting for readers.
I have only a few questions to the methods and results sections.
I hope that my comments have been helpful and valuable. Detailed comments below.

The authors wrote:
<<Specifically, six categories based on the collection months of the host species were used: “March-April” (39 samples), “May-June” (36 samples), “July” (47 Samples), “August” (61 samples), “September” (82 samples), and “October-November” (15 samples). The samples were also divided into growth habitat categories including “coniferous forests” (30 samples), “hardwood forests” (109 samples), “mixed forests” (131 samples), and “scrub forests” (10 samples).>>

Please, add map with locations (and coordinates) for the study sites. Moreover, please add the minimal distance between the study sites (altogether 280 sites), to avoid the pseudoreplication effect ( more : https://web.ma.utexas.edu/users/mks/statmistakes/pseudorep.html ). Study sites (i.e. geographical location) strongly influence ECM fungal biota, even if the habitat (type of forest ecosystem) is the same; it concerns both ECM fungi (Leski et al. 2019, Biological Conservation: https://doi.org/10.1016/j.biocon.2019.108206), saprotrophic fungi, and/or pathotrophic fungi (Rudawska et al. 2022, Forest Ecology and Management: https://doi.org/10.1016/j.foreco.2022.120274).

Please, describe study sites and please, list the tree species at coniferous, hardwood (I suppose - decidous), mixed, and scrub forest habitats, with special attention to ECM trees (most tree species of Pinaceae and Fagales).

Some macrofungal species, such as genera Helvella, Russula, Suillus, Xerocomus, Lactifluus, Laccaria, and Inocybe form ECM symbiosis with trees. ECM symbiosis is obliatory interaction for both, ECM fungi and ECM trees. So the data on distribution of ECM fungi and their tree partners is important in the context of fungicolous microfungal species.

Please, re-analyze the data with focus on ECM fungal species, and their ECM tree partners.
In addition, please describe precisely the study sites. For now, it is insufficiently described.

I'm not sure, that only two seasons (2020, 2021) is sufficient for such study. If I remember correctly, the proper time for sporocarp surveying is 4-5 years, or ever 7 years or longer. In my work, I'm focused on the metagenomic analyzes of soil and roots, not fungal sporocarp surveing. Nevertheless, one season can be largely different from another, what makes 4-5 year old sporocarp surveing (or longer) justified. Please look at Body et al. 2014, Fungal Ecology:  https://doi.org/10.1016/j.funeco.2013.10.006 ).

Please, add year (2020 vs 2021), altitude, tree partners, study sites characteristic, and other such elements to the results section. For now, seasons/months are well represented, but the precipitation and temperature in exact month in exact year makes the sporocarp production possible, or not. For example, in European forests in the late July, we can found numerous ECM fungal sporocarps of different ECM  fungal genera, or almost no ECM fungi, like in this year (because of drought, high temperature, and ongoing climate change).

Data on the weather conditions (precipitation, temperature) would also be helpful for better understanding of your results.

Author Response

Comments 1: Please, add map with locations (and coordinates) for the study sites. Moreover, please add the minimal distance between the study sites (altogether 280 sites), to avoid the pseudoreplication effect ( more : https://web.ma.utexas.edu/users/mks/statmistakes/pseudorep.html ). Study sites (i.e. geographical location) strongly influence ECM fungal biota, even if the habitat (type of forest ecosystem) is the same; it concerns both ECM fungi (Leski et al. 2019, Biological Conservation: https://doi.org/10.1016/j.biocon.2019.108206), saprotrophic fungi, and/or pathotrophic fungi (Rudawska et al. 2022, Forest Ecology and Management: https://doi.org/10.1016/j.foreco.2022.120274).

Please, describe study sites and please, list the tree species at coniferous, hardwood (I suppose - decidous), mixed, and scrub forest habitats, with special attention to ECM trees (most tree species of Pinaceae and Fagales).

Response 1: Thank you for pointing it out. Given that our study sites encompass regions with confidentiality restrictions, in compliance with pertinent legal statutes and our verbal accord with the local forestry administration, we are precluded from divulging details regarding the geography and climate of these zones. Consequently, we cannot supply detailed information such as maps, coordinates, and climatic data for the study sites. Nonetheless, within the confines of legal permissions, we have presented a general profile of the study sites' geography, vegetation, and climate within our publication. Furthermore, the minimum distances among the study sites and the dominant tree species within each habitat have been listed. See lines 90 to 114 of revised manuscript or the following paragraphs.

“Samples were collected from the Qinling Mountain within the Shaanxi Province (China) from March 2020 to November 2021. The sampling regions encompassed a range of areas, including Ningshan, Zhashui, Mei, Foping, Taibai, and Chang'an districts. The survey extended from its lowest elevation at Longxu Gully in Xialiang Town, Zhashui County (at 608 meters) to its highest in the Pinghe Liang area, Ningshan County (at 2364 meters), showcasing a substantial altitudinal range of 1756 meters. The most southerly sampling points were near Ningshan County, Shaanxi Province (33.38 N, 108.26 E), and the northernmost at Lintong District, Shaanxi Province (34.34 N, 109.26 E), while the most easterly sampling point was at Lintong District, Shaanxi Province (34.34 N, 109.26 E), and the westernmost was near Taibai County, Shaanxi Province (34.15 N, 107.26 E).

The sampling region is located within the climatic transition belt from subtropical to warm temperate zones, marking it as one of the primary areas rich in macrofungal diversity in the Qinling Mountains. The vegetation in this region comprises various forest ecosystems, including decidous broadleaf forests, mixed coniferous and broadleaf forests, coniferous forests, and scrub forests. The deciduous broadleaf forests are dominated by species such as Quercus variabilis, Quercus acutissima, and Quercus wutaishanica. The mixed coniferous and broadleaf forests feature Pinus armandii, Quercus wutaishanica, and Quercus acutissima as their principal tree species. The coniferous forests are characterized predominantly by Pinus tabulaeformis, Pinus armandii, and Metasequoia glyptostroboides. Additionally, the scrub forests are notably dominated by Fargesia spathacea.

To avoid the pseudoreplication effect, a minimum elevation gradient of over 100 meters was maintained across all sampling sites. To minimize the influence of factors such as humidity on the experimental outcomes and to ensure the acquisition of a sufficient quantity of samples, the collection was scheduled within the 1 to 3 days following rainfall at the pre-designated sampling sites.”

Comments 2: Some macrofungal species, such as genera Helvella, Russula, Suillus, Xerocomus, Lactifluus, Laccaria, and Inocybe form ECM symbiosis with trees. ECM symbiosis is obliatory interaction for both, ECM fungi and ECM trees. So the data on distribution of ECM fungi and their tree partners is important in the context of fungicolous microfungal species. Please, re-analyze the data with focus on ECM fungal species, and their ECM tree partners. In addition, please describe precisely the study sites. For now, it is insufficiently described.

Response 2: Thank you for pointing it out. EMC fungi typically refer to Ectomycorrhizal (ECM) fungi, which form symbiotic relationships with many trees and play a key role in the nutrient and carbon cycles of forest ecosystems. Our study primarily focused on fungicolous fungi. Fungicolous fungi mainly refer to fungi that reside within other fungi, particularly within the fruiting bodies of macrofungi. There is a clear distinction between ECM fungi and fungicolous fungi; therefore, incorporating analysis and discussion of ECM fungi in the paper could impact the logic and integrity of the existing work. Nonetheless, ECM fungi are very important, and we will focus on them in our upcoming research. We greatly appreciate your valuable suggestions.

Comments 3: I'm not sure, that only two seasons (2020, 2021) is sufficient for such study. If I remember correctly, the proper time for sporocarp surveying is 4-5 years, or ever 7 years or longer. In my work, I'm focused on the metagenomic analyzes of soil and roots, not fungal sporocarp surveing. Nevertheless, one season can be largely different from another, what makes 4-5 year old sporocarp surveing (or longer) justified. Please look at Body et al. 2014, Fungal Ecology:  https://doi.org/10.1016/j.funeco.2013.10.006).

Response 3: Thank you for pointing it out. Our team has been conducting a multi-year, continuous survey of the macrofungal resources in the Qinling region since 2008 and is well-acquainted with the geography, climate, and distribution of macrofungi there. For instance, we are very clear about which types of macrofungi appear in which parts of the Qinling and during which seasons of the year. Therefore, when designing the sampling sites, we take into account various factors, including climate, to ensure that a two-year sampling period is sufficient for drawing experimental conclusions. Hence, we believe that a two-year sampling period is adequate for the conclusions of the current experiment. Of course, a longer experimental period would certainly be better and likely yield additional findings. We greatly appreciate your insightful recommendations and will persist in carrying out longer-term associated research in our subsequent investigations.

Comments 4: Please, add year (2020 vs 2021), altitude, tree partners, study sites characteristic, and other such elements to the results section. For now, seasons/months are well represented, but the precipitation and temperature in exact month in exact year makes the sporocarp production possible, or not. For example, in European forests in the late July, we can found numerous ECM fungal sporocarps of different ECM  fungal genera, or almost no ECM fungi, like in this year (because of drought, high temperature, and ongoing climate change). Data on the weather conditions (precipitation, temperature) would also be helpful for better understanding of your results.

Response 4: Thank you for pointing it out. As previously stated in my initial response, we are constrained by confidentiality agreements from revealing detailed information regarding the climate and geographical specifics of our sampling sites. Consequently, for our forthcoming studies, we intend to focus on areas within the Qinling region that are not subject to such restrictions. We extend our gratitude for your insightful recommendations.

Round 2

Reviewer 1 Report

This manuscript is a novel and original study that explores the diversity, composition, and trophic modes of fungicolous fungal communities residing in host macrofungi in the Qinling Mountains, China. Additionally, the authors provide valuable information related to the quantification and comparison of carbohydrate-active enzymes (CAZymes) encoded by saprophytic, pathogenic, and symbiotic fungi inhabiting macrofungi. The primary strength of the manuscript lies in the potential for fungicolous fungi to serve as model organisms, facilitating a better understanding of species interactions, fungal evolution and divergence, and fungicolous mechanisms.

The authors have attended the observations made to their original draft, then my opinion is that now is ready to be published.

This manuscript is a novel and original study that explores the diversity, composition, and trophic modes of fungicolous fungal communities residing in host macrofungi in the Qinling Mountains, China. Additionally, the authors provide valuable information related to the quantification and comparison of carbohydrate-active enzymes (CAZymes) encoded by saprophytic, pathogenic, and symbiotic fungi inhabiting macrofungi. The primary strength of the manuscript lies in the potential for fungicolous fungi to serve as model organisms, facilitating a better understanding of species interactions, fungal evolution and divergence, and fungicolous mechanisms.

The authors have attended the observations made to their original draft, then my opinion is that now is ready to be published.